# Role of the Innate Immune Response in Glomerular Disease Pathogenesis: Focus on Podocytes

**DOI:** 10.3390/cells13131157

**Published:** 2024-07-06

**Authors:** Wadih Issa, Rachel Njeim, Arianna Carrazco, George W. Burke, Alla Mitrofanova

**Affiliations:** 1Department of Internal Medicine, Saint Joseph University, Beirut 1107 2180, Lebanon; 2Katz Family Division of Nephrology and Hypertension, Department of Medicine, University of Miami Miller School of Medicine, Miami, FL 33136, USA; 3Peggy and Harold Katz Family Drug Discovery Center, University of Miami Miller School of Medicine, Miami, FL 33136, USA; 4Division of Kidney-Pancreas Transplantation, Department of Surgery, Miami Transplant Institute, University of Miami Miller School of Medicine, Miami, FL 33136, USA

**Keywords:** podocyte, inflammation, innate immunity, TLRs, STING, glomerular disease

## Abstract

Accumulating evidence indicates that inflammatory and immunologic processes play a significant role in the development and progression of glomerular diseases. Podocytes, the terminally differentiated epithelial cells, are crucial for maintaining the integrity of the glomerular filtration barrier. Once injured, podocytes cannot regenerate, leading to progressive proteinuric glomerular diseases. However, emerging evidence suggests that podocytes not only maintain the glomerular filtration barrier and are important targets of immune responses but also exhibit many features of immune-like cells, where they are involved in the modulation of the activity of innate and adaptive immunity. This dual role of podocytes may lead to the discovery and development of new therapeutic targets for treating glomerular diseases. This review aims to provide an overview of the innate immunity mechanisms involved in podocyte injury and the progression of proteinuric glomerular diseases.

## 1. Introduction

The glomerulus is a specialized network of capillaries located at the beginning of a nephron with an important function in the filtration of blood, removing waste products and the formation of glomerular filtrate, which moves through the renal tubule and further forms into urine. The glomerular filtration barrier includes endothelial cells, the glomerular basement membrane (GBM) and podocytes. Abnormalities in the glomerular filtration barrier integrity leads to protein loss into the urine. Approximately 80% of individuals who develop end-stage kidney disease have proteinuric kidney disease resulting from glomerular dysfunction [1].

Podocytes, being highly specialized epithelial cells of the glomerulus, are essential for the selective permeability of the glomerular filtration barrier, helping to ensure that essential molecules like proteins remain in blood. Besides that, podocytes also have a very important role in the synthesis and repair (together with endothelial cells) of the glomerular filtration barrier [2,3] and the the production of VEGF [4,5] and platelet-derived growth factor (PDGF) [6]. Interestingly, podocytes have been found to express various pattern recognition receptors (PRRs), among which C-type lectin receptors (CLRs), Toll-like receptors (TLRs), NOD-like receptors (NLRs), RIG-I-like receptors (RLRs) as well as intracellular DNA sensors such as the cyclic GMP-AMP synthase-stimulator of the interferon genes pathway and inflammasome signaling are recognized. Traditionally, the immune response and podocytes are thought of together in the context of lupus nephritis or IgA nephropathy, which was extensively studied and reviewed somewhere else [7,8,9,10,11,12]. However, recent findings also suggest that other glomerular diseases such as diabetic kidney disease, renal injury associated with Alport Syndrome, membranous nephropathy, minimal change disease and focal segmental glomerulosclerosis may be considered as immune-mediated diseases characterized by inflammation. However, the intricate mechanisms of immune response associated with podocytes injury remain poorly understood. The present review summarizes our current knowledge on the immune mechanisms triggering podocyte injury and determines if different mechanisms are affected in different glomerular diseases.

## 2. Podocytes in Innate Immunity

### 2.1. Toll-like Receptors (TLRs) Signaling

The TLR family consists of ten members in humans (TLR1-TLR10) and twelve members in mice (TLR1-TLR9 and TLR11-TLR13). TLRs are single-pass transmembrane receptors located on the cell surface or within endosomal compartments (TLR3, TLR7, TLR8 and TRL9) and are composed of three main domains: (1) an N-terminal domain (NTD) that recognizes pathogen-associated molecular patterns (PAMPs) and damage-associated molecular patterns (DAMPs), (2) a middle single helix transmembrane domain (TMD) that traverses the membrane and (3) a C-terminal domain (CTD) [13] that initiates downstream signal transduction through its toll-IL-1 receptor (TIR) homologous domain [14,15,16]. TLRs are essential for innate immune defense against invading microorganisms and contribute to sterile inflammation in various diseases, including diabetes. Notably, both human and murine podocytes express all TLRs, although the expression levels vary among different TLRs.

Exposure to high glucose is associated with direct TLR4 activation and contributes to podocyte damage and interstitial fibrosis progression [17]. Additionally, the constitutive expression of TLR4 in podocytes was shown to also be upregulated in membranoproliferative glomerulonephritis (MPGN) [18]. Moreover, early studies demonstrated that both naked and protein-bound mitochondrial DNA (mtDNA) can activate TLR9 and the advanced glycation end product-specific receptor (RAGE) [19,20]. De novo TLR9 expression in the podocytes of some patients with glomerular diseases was found [21,22,23,24], suggesting that mtDNA may also facilitate podocyte apoptosis [25]. Interestingly, another study has also shown that injured podocytes have TLR9 overexpression associated with the development of MPGN in a mouse model of BXSB/MpJ-Yaa [26]. Thus, TLRs may represent potential therapeutic targets in different glomerulopathies (Table 1).

### 2.2. Nucleotide-Binding Oligomerization Domain (NOD)-like Receptors (NLRs) Signaling

NLRs belong to the innate receptors, primarily located in the cytoplasm, while some are also presented in mitochondria. There are 22 identified NLR genes in humans and 35 in mice [27,28]. NLRs have a conserved tripartite structure consisting of (1) a C-terminal leucine-rich repeat (LRR) domain that acts as a ligand detector, (2) a central nucleotide-binding (NACHT) domain that binds nucleotides and forms oligomers and (3) an N-terminal effector domain, such as CARD, pyrin domain (PYD) or baculoviral inhibitory repeat (BIR)-like domain, which facilitates interactions with downstream proteins [28,29]. NLRs are categorized into four subfamilies based on their N-terminal domains: NLRAs, NLRBs (or NAIPs), NLRCs and NLRPs. The NLRA subfamily, represented solely by CIITA, contains an acidic transactivating domain. NLRBs, such as NAIP in humans and NAIP1-7 in mice, are characterized by an N-terminal BIR-like domain. NLRCs include NOD1, NOD2 and NLRC3-5, each featuring a CARD domain, while NLRX1 is located in the mitochondrial matrix. NLRPs comprise 14 members (NLRP1-14), all possessing PYD, NACHT and LRR domains, with the exception of NLRP10, which lacks an LRR domain. In animals, NAIPs and NLRC4 can combine to form inflammasomes [30,31]. Most NLRs play a role in pathogen/damage sensing, antigen presentation, inflammasome activation or inflammatory signaling inhibition (reviewed in detail elsewhere [32,33]). Different NLRs recognize different PAMPs and DAMPs (Table 1). NLRC3 [34], NLRC5 [35], NLRX1 [36] and NLRP6 [37] are implicated in dampening inflammation, whereas NLRP3, NLRP1, NLRP2, NLRP6 and NLRP7 are capable of forming inflammasomes and exacerbating the immune system response [38,39,40,41,42]. NLRP3 is traditionally associated with canonical inflammasome activation, converting procaspase-1 into caspase 1 and leading to IL-1β/IL-18 secretion and the activation of pyroptosis. However, in a noncanonical inflammasome-independent manner, NLRP3 activates procaspase-11 [43].

Recent research indicates that several NLRs are significant contributors to various kidney diseases. Studies have shown that high glucose-induced NADPH oxidase activation, mediated by thioredoxin-interacting protein (TXNIP), triggers NLRP3 inflammasome activation in podocytes, exacerbating podocyte injury [44]. Notably, treatment with ophiocordyceps sinensis, an artificial agent, has been effective in the downregulation of P2X7R and NLRP3 expression, thereby ameliorating podocyte injury [45]. NOD2 has been reported to be expressed in glomerular mesangial cells, podocytes, endothelial cells, tubular epithelial cells, and inflammatory cells. NOD2 deficiency can reduce the downregulation of nephrin induced by high glucose, thereby maintaining podocyte integrity [46].

### 2.3. C-Type Lectin Receptors (CLRs) Signaling

CLRs are transmembrane pattern recognition receptors, expressed by most cell types but mainly by dendric and myeloid cells (reviewed in [47]). CLR family proteins have one (or more) domain similar to carbohydrate recognition domains (CRDs), although they do not necessarily bind to carbohydrate structures. Upon pathogen binding, CLRs trigger distinct signaling pathways leading to the activation of NF-κB, type-I interferon and inflammasomes (reviewed in [48]), which induces specific cytokines expression and determines the T cell polarization fate.

CLRs can be categorized into two groups: (1) those that belong to the mannose receptor family and (2) those that belong to the asialoglycoprotein receptor family. CLRs from the first group include the mannose receptor (CD206) and DEC205 (CD205). CLRs from the second group include the DC-SIGN (CD209), Langerin (CD207), MGL (CD301) and CLEC5A (MDL1) receptors and include the DC-associated C-type lectin 1 (Dectin 1) and DC immunoreceptor (DCIR) subfamilies. In turn, the Dectin 1 subfamily includes the Dectin 1 (CLEC7A), MICL (CLEC12A), CLEC2, DNGR1 and CLEC12B receptors, while the DCIR subfamily contains the Dectin 2, BDCA2 (CD303), Mincle (CLEC4E) and DCIR (CLEC4A) receptors (reviewed in [49]).

CLRs bind to ligands in a calcium-dependent manner through the recognition of mannose, fucose and glucan carbohydrate structures using CRDs [50]. CLRs can trigger signaling pathways through immunoreceptor tyrosine-based activation motif (ITAM)-containing adaptor molecules like FcRγ or DAP12 [51,52,53,54]. They can also activate signaling pathways via protein kinases and phosphatases that directly or indirectly interact with their cytoplasmic domains [55,56,57,58]. Some CLRs, such as DC-SIGN, BDCA2, DCIR and MICL, can induce a signaling pathway that modulates TLRs at the transcriptional and post-transcriptional levels (reviewed in [49]).

Notably, podocytes express some CLRs, that have been shown to play a role in the pathogenesis of glomerular diseases. Thus, DC-SIGN is shown to be overexpressed in the glomeruli of a mouse model of lupus nephritis, while its inhibition in vivo diminished proteinuria and led to renal function improvement [59]. Moreover, the same study demonstrated that podocytes treatment with serum from patients with lupus nephritis results in increased DC-SIGN expression. Additionally, podocytes have been shown to induce cytoskeleton re-arrangement in response to treatment with CLEC2, the CLR highly expressed by platelets, via the dissociation of F-actin from podoplanin, the endogenous ligand of CLEC2, which is most intensely expressed in podocytes [60]. CLEC14A expression is shown to be associated with podocyte injury in the Adriamycin-induced mouse model of focal segmental glomerulosclerosis [61], as discussed below.

### 2.4. Cyclic GMP-AMP (cGAS)-Stimulator of Interferon Genes (STING) Signaling

The cGAS-STING signaling pathway initiates a cascade of regulatory mechanisms involved in autoinflammatory, autoimmune, autophagy, apoptosis and degenerative diseases. cGAS detects the presence of double-stranded DNA in the cytosol, leading to the production of the second messenger cyclic GMP-AMP (cGAMP) [62,63,64] and STING activation, which subsequently triggers innate and adaptive immune responses [65]. While it is commonly believed that STING is primarily localized at the outer membrane of the endoplasmic reticulum, some studies have also reported its presence in the mitochondrial membrane [66,67].

STING contains a four-span transmembrane cytosolic N-terminal domain, a cytosolic ligand-binding domain whose C-terminal tail is protruding from the ER membrane and a connector region allowing for anchorage to the ER [62,68]. The protrusion of the C-terminal tail allows cGAMP to bind and activate STING. Upon binding to STING, the cGAMP-STING complex will undergo a conformational change, forming a β-sheet that covers the ligand-binding domain [64]. The activation of STING will cause the phosphorylation of STING itself as well as the TANK-binding kinase 1 (TBK1) and interferon regulatory factor 3 (IRF3) transcription factor [69]. Once phosphorylated IRF3 dimerizes and traffics to the nucleus, this initiates the production of type 1 interferons, resulting in the activation of multiple cellular functions such as pro-inflammatory cytokine production, apoptosis and autophagy [70]. Importantly, activated STING has been shown to recruit other kinases, such as mitogen-activated protein kinase kinase kinase 14 (MAP3K14 or NIK) and heterotrimeric IκB kinase (IKK). This recruitment promotes non-canonical nuclear factor kappa B (NF-κB) and canonical NF-κB activation, respectively, leading to a wide range of effects. [71,72,73].

Originally, the cGAS-STING pathway was discovered as the immune system’s ability to detect and respond to DNA in the cytoplasm of cells in the context of infections and cancer [74,75,76]. However, it is clear now that the STING pathway plays a crucial role in the self-DNA detection released from damaged mitochondria, dying cells or tumor cells. Thus, the cGAS-STING pathway has been shown to regulate inflammation levels and the ehomeostasis of energy in obesity [77,78], in systemic lupus erythematosus [79,80], in tubular epithelial cells in a mouse model of chronic kidney disease [81] and in acute kidney injury [82,83]. STING has also a role in the pathogenesis of insulin resistance and chronic low-grade inflammation observed in obesity [84]. Additionally, diabetes-associated lipotoxicity is known to trigger mitochondrial damage and mitochondrial DNA (mtDNA) leakage, which results in STING activation in podocytes (Table 1) at early stages of the renal disease [85], including db/db mice and mice on a high-fat diet [86]. Another study reported that in human podocytes with APOL1 risk, the activation of the alleles G1 and G2 STING is noted, leading to IFN signaling upregulation [87]. Therefore, targeting STING may be a good option for the treatment of patients with chronic kidney disease.

### 2.5. Retinoic Acid-Inducible Gene (RIG)-I-like Receptors (RLRs) Signaling

RLRs are cytoplasmic RNA helicases with a DExD/H box, which allows them to detect viral or self-RNA, mediating the induction of type I interferons [88,89]. So far, the RLRs family of receptors consists of three members: RIG-I, melanoma differentiation-associated protein 5 (MDA5) and laboratory of genetics and physiology 2 (LGP2). RIG-I and MDA5 have two amino-terminal caspase activation and recruitment domains (CARDs) responsible for downstream signal transduction. In contrast, LGP2 lacks CARDs and is commonly believed to modulate RIG-I and MDA5 activities [90,91]. Following RNA binding and oligomerization, RLRs interact with the CARD domain of the mitochondrial antiviral-signaling protein (MAVS), which subsequently triggers the activation of TANK-binding kinase 1 (TBK1) and IκB kinase-ε (IKKε), further activating interferon regulatory factor 3 (IRF3) and IRF7 and leading to the transcriptional activation of interferon (IFN) and proinflammatory cytokine genes [88,89,92].

The RLRs pathway is activated by foreign, altered or ectopic RNA [93]. Thus, RIG-I identifies viral genomic RNAs primarily through their 5ʹ-triphosphate signature [94,95]. In addition to the 5′-triphosphate structure, the 5′-diphosphate moiety can also be recognized by RIG-I [96]. The activation of RIG-I by host RNAs can be triggered by cellular non-coding RNAs, such as 5S ribosomal RNA pseudogene 141 (RNA5SP141) [97] or an endogenous RNA known as “vault RNA” transcribed by RNA polymerase III [98]. MDA5 can also be activated by host RNAs. For instance, the release of mitochondrial RNA (mtRNA) into cytosol in a state of reduced polyribonucleotide nucleotidyltransferase 1 (PNPT1) expression leads to MDA5 activation [99], while BAX-BAK1-mediated mtRNA release does not activate MDA5 but rather RIG-I instead [100]. In a case of a loss of function of helicase SUV3 and the polynucleotide phosphorylase PNPase, mtRNA is accumulated in the cytoplasm and leads to MDA5-mediated IFN production [99]. Nevertheless, the mechanisms associated with the differential activation of MDA5 versus RIG-I by mtRNA species have yet to be elucidated.

A crucial role of RIG-I and MDA5 in responses associated with kidney diseases has been reported (Table 1). Thus, RIG-I signaling is upregulated in proximal tubules in ischemic acute kidney injury [101], in patients with IgA nephropathy [102], in models of unilateral ureteral obstruction (UUO) and in patients with a moderate degree of renal fibrosis [103]. Similarly, patients with severe lupus nephritis or proteinuric IgA nephropathy have high levels of MDA5 expression in glomeruli, mainly in mesangial cells [104]. Despite these advances, the role of RLRs in podocytes remains poorly understood. Thus, RIG-I signaling in podocytes is associated with structural and functional changes, which, in turn, may result in increased glomerular permeability to serum proteins and/or glomerular responses to injury [105]. The treatment of podocytes with PolyIC, a synthetic dsDNA, induced the expression of RIG-I and MDA5 in a dose- and time-dependent manner, which led to an increase in IFN-β and interleukin-6 (IL-6) expression, as well as podocyte actin reorganization [106]. Interestingly, Apolipoprotein L1-induced glomerular damage was reduced in adeno-associated virus (AAV)-shRIG-I mice [107]. Additionally, RIG-I activation was shown to lead to increased albumin permeability in glomerular mesangial cells [108] and an RIG-I-dependent increase in chemokine ligand 5 (CCL5) and C-X-C motif chemokine ligand 10 (CXCL10) in mesangial cells [104,109]. However, further research is warranted to better elucidate the role of RLRs in glomerular cells’ physiology and pathophysiology.

## 3. Immune Response in Podocytes from Different Glomerular Diseases

### 3.1. Diabetic Kidney Disease (DKD)

DKD remains the leading cause of chronic kidney disease globally, where up to 40% of patients with diabetes develop progressive nephropathy resulting in end-stage kidney disease. The pathogenesis of DKD involves multiple mechanisms and remains incompletely understood, which results in the lack of specific therapies. Historically, DKD is considered as a “non-inflammatory” disease, but more evidence from experimental and clinical research clearly demonstrates that innate immune responses are significantly involved in DKD development and progression [110,111].

Among different TLRs, the expression of TLR2, TLR4, TLR5, TLR7, TLR8 and TLR9 has been reported in DKD, while TLR2 and TLR4 are the most extensively studied receptors (reviewed in [112]). Enhanced glomerular TLR4 expression is observed in the streptozotocin-induced DKD mouse model, whereas mice with TLR4 deletion are protected against disease development [17]. Limited research on TLR3 and TLR9 receptors in DKD indicates that both receptors are activated in the kidney of the Apolipoprotein E knockout streptozotocin-induced mouse model [113]. An increased expression of TLR3 has also been observed in tubules from patients with DKD [114]. However, there are no data connecting the TLR3 and TLR9 activation with mtDNA release into the cytosol in DKD, which may warrant future investigation.

Not many data are available regarding the role of RIG-I and MDA5 in DKD. Thus, an association of RIG-I/MDA5 and the interferon alpha beta gene set in patients with type 1 diabetes was found using genome-wide association study (GWAS) analysis [115]. Interestingly, an important role of RIG-I/MAVS signaling was reported in the pancreas and is associated with diabetes progression. For example, elevated RIG-I in type 2 diabetes inhibits pancreatic β cell mass in a Src/STAT3-dependent manner [116]. Similarly, in human fulminant type 1 diabetes, the overexpression of RIG-I and MDA5 in the β cell was shown, which led to β cell death [117]. A recent study demonstrated that RLR signaling shifts energy metabolism from glycolysis to the hexosamine biosynthetic and pentose phosphate pathways through MAVS [118], suggesting an important RLR signaling role in metabolic disorders.

Emerging evidence underscores the upregulation of inflammasome-associated factors like NLRP3, IL-18 and IL-1β in patients and in mouse models of DKD [119]. The inhibition of the NLRP3 inflammasome has been found to reduce podocyte damage by attenuating lipid accumulation in DKD [120]. Additionally, the purinergic receptor P2X7R has been implicated in mediating renal inflammation and injury in diabetic nephropathy through NLRP3 inflammasome activation. Notably, treatment with ophiocordyceps sinensis, a parasitic fungus, which is highly prized in traditional Chinese medicine, has been effective in downregulating the expression of P2X7R and NLRP3, thereby ameliorating podocyte injury [45]. Interestingly, a recent study reported that the protective effects of semaglutide and liraglutide on podocytes in DKD could be achieved via the regulation of the NLRP3 inflammasome pathway and the inhibition of the pyroptosis-related genes expression and inflammatory factors IL-1β and IL-18 [121]. Similarly, dapagliflozin was found to alleviate pyroptosis in podocytes via the regulation of the heme oxygenase 1 (HO-1)/NLRP3 axis [122]. In DKD, it was shown that podocyte-specific Nlrp3 or Caspase-1 deficiency provided protection from disease progression [123]. In contrast, another study found that the NLRP3-specific inhibitor, MCC950, does not have protective effects on the kidney in streptozotocin-induced diabetic mice, as it did not show any improvements in renal inflammation, mesangial expansion and glomerulosclerosis [124]. In contrast, another report showed a renoprotective effect of MCC950 in a model of oxalate nephropathy [125].

Based on our previous report, increased STING phosphorylation is associated with kidney disease progression in the db/db mouse model of DKD, while pharmacological STING inhibition is shown to protect from DKD progression [126,127]. Importantly, STING activation alone led to proteinuria and podocyte foot process effacement in wildtype mice [126,127]. Using the eNOS;db/db mouse model of DKD and DKD rats, an increased activity of the cGAS-STING pathway was also observed [128]. Somewhat surprisingly, in a diabetic condition, the activation of the STING pathway was evidenced in glomerular endothelial cells, which leads to glomeruli damage and, ultimately, podocyte loss [108,129,130].

While no literature supports the role of CLRs in DKD, there is a scientific report that demonstrates the involvement of CLRs in other diabetes-associated complications. Thus, increased Dectin-1 expression was shown in the cardiac tissue of diabetic mice, where the authors suggested that Dectin-1 aggravates diabetic cardiomyopathy via the activation of Syk/NF-κB signaling in response to high glucose levels [131].

Thus, targeting innate immune signaling molecules may be beneficial in protecting kidney tissue and podocytes and preventing DKD progression. However, more research in this area is needed.

### 3.2. Alport Syndrome (AS)

AS is a rare genetic disease associated with mutation in alpha chains of the collagen type IV (COL4) gene. In about 80% of cases, AS is an inherited disease associated with an X-linked pattern and caused by collagen type IV gene mutations [132]. In total, 1 in 50,000 newborns is affected by AS, and males are more likely to display symptoms compared to females. In the US, the incidence of end-stage kidney disease (ESKD) is about 3% and 0.2% in children and in adults, respectively [133]. The impaired production of the COL4 α345 network in the glomerular basement membrane leads to injury via the activation of adhesion kinases in podocytes [134], the overexpression of endothelin receptors [135], glomerular inflammation [136], tubulointerstitial fibrosis and ESKD. Histopathology in AS includes focal and segmental glomerulosclerosis and interstitial fibrosis, tubular atrophy, the presence of lymphocytes and plasma cells and multi-lamellation of the GBM lamina densa. Given the association between collagen type IV abnormalities and chronic inflammation, there is renewed interest in the development of new anti-inflammatory therapies in AS.

Chronic kidney inflammation is known to include the activation of a proinflammatory response in the resident kidney cells such as endothelial and mesangial cells, podocytes and tubular epithelial cells. Cytokines produced by the resident kidney cells, in turn, attract macrophages to further damage the tissue. Thus, in the COL4α3 mouse model of AS, increased tumor necrosis factor α (TNFα) mRNA in the mesangium and podocytes was reported as the disease progressed [137]. High levels of transforming growth factor β (TGFβ) and high mobility group box 1 (HMGB1) were reported in pediatric patients with AS compared to healthy controls and negatively correlated with the estimated glomerular filtration rate (eGFR) [138], suggesting that inflammatory mechanisms are triggered before the onset of proteinuria. Further, another study showed that the levels of monocyte chemoattractant protein 1 (MCP-1), the regulator of monocyte infiltration, were significantly higher in the urine of patients with AS. Moreover, in the 28 patients followed for 5 years, urinary MCP-1 also negatively correlated with the slope of eGFR decline [139]. Early inflammation events in AS are thought to be regulated by the nuclear factor κ-light-chain-enhancer of activated B cells (NF-κB). Indeed, using the collagen type IV alpha 5 (COL4α5) mouse model of AS with the podocyte-specific deletion of p53, a regulator of NF-κB repression, by the glucocorticoid receptor [140] was associated with enhanced renal dysfunction, foot process effacement and renal inflammation [141]. Interestingly, a recent report suggests that using an activator of nuclear factor erythroid 2-related factor 2 (Nrf2) in the COL4α5 mouse model of AS showed the amelioration of inflammation, as well as fibrosis and glomerulosclerosis, in association with an extended lifespan [142].

Our own studies also demonstrated the activation of STING in the glomeruli of the collagen type IV alpha 3 (COL4α3) mouse model of AS [127]. Importantly, the pharmacological STING inhibition in AS mice resulted in reduced proteinuria, improved renal function and an increase in the lifespan of mice. Interestingly, a recent study using an AS transgenic mouse model created by transferring a pathogenic mutation COL4α3 p.Gly801Arg, the equivalent of the COL4α3 mutation G801R, which is located two residues earlier, showed persistent ER stress and MyD88/p38 MAPK signaling activation, leading to inflammation and apoptosis injury in the kidney [143].

Thus, targeting immune pathways may represent another avenue in the development of new therapeutic strategies for managing renal failure in AS.

### 3.3. Membranous Nephropathy (MN)

MN is another kidney disease that may affect individuals across different age groups. Among the pathological characteristics of MN, we can indicate the deposition of immune complexes on the outer side of the glomerular basement membrane (GBM) and diffuse GBM thickening. Two types of MN are identified: (1) primary MN with no known cause (up to 30% of cases) and (2) secondary MN, which is caused by systemic autoimmune diseases, drugs, infections, hematopoietic stem cell transplantation or malignancy [144]. An immunofluorescence analysis of primary MN depicts granular deposits of IgG (predominantly IgG4) and C3 complement along the capillary wall (reviewed in [145]).

Recent progress in understanding of primary MN pathogenesis revealed that podocyte express protein phospholipase A2 receptor (PLA2R) serves as an antibody marker in 70% of patients with primary MN [146,147]. Five single-nucleotide polymorphisms (SNPs) of PLA2R (*rs4664308*, *rs3749119*, *rs3749117*, *rs3828323* and *rs2187668*) have been found to be associated with primary MN in an Indian cohort [148]. Interestingly, PLA2R-negative patients with MN demonstrated a high expression of other antigens, such as neural epidermal growth factor-like 1 protein (NELL-1) [149], protocadherin 7 (PCDH7) [150] and neural cell adhesion molecule 1 (NCAM1) [151], which were not found to be expressed by podocytes. Importantly, the role of innate immunity in MN was established in a Taiwan population, where the association of IL-6, nephrin, TLR4 and TLR9 genes with primary MN was shown [152]. In the case of the TLR4 gene, the SNPs’ (*rs10983755* A/G, *rs1927914* A/G, *rs10759932* C/T, *rs11536889* C/G) presence has been demonstrated in MN patients compared to control patients. In addition, microarray data from patients with MN showed 574 differentially expressed genes compared to controls, where several important genes (*ZYX*, *CD151*, *N4BP2L2-IT2*, *TAPBP*, *FRAS1* and *SCARNA9*) associated with the actin cytoskeleton, GBM function and antigen presentation can serve as novel markers of MN [153].

Thus, the pathogenesis of MN is highly complex, which requires the further investigation of the mechanisms of podocyte immunity, aiming at the discovery of new effective therapeutic targets.

### 3.4. Minimal Change Disease (MCD)

MCD is a type of nephrotic syndrome, which is very common in children and adolescents, with up to 16% cases only in adults [154]. Usually, MCD is characterized by normal histology and extensive podocyte foot process effacement on electron microscopy analysis (reviewed in [155]). Podocyte injury in MCD may be idiopathic, genetic or reactive, with steroids as a first-line therapy. The idiopathic and reactive MCD are steroid-sensitive forms, while genetic MCD is a steroid-resistant form with worse renal outcomes. The sudden-onset proteinuria is a clinical hallmark of MCD. Currently, the mechanisms of proteinuria in MCD are poorly understood (reviewed in [156]). Despite this, some research findings suggest an association of immune system dysregulation and MCD progression.

Thus, the lack of immune complexes or inflammatory cells glomeruli in patients with MCD led to the idea that circulating factors are the main triggers of proteinuria and podocyte injury. Thus, various cytokines (such as interleukin (IL)-13 [157], IL-8 [158], IL-4 [159]) released by T cells may lead to MCD development. In addition, some studies have reported the involvement of B cells. Among them, increased serum levels of the B cell-activating factor in patients with MCD have been shown [160], which may lead to T cells activation. Interestingly, a recent study also demonstrated the presence of specific autoantibodies that target nephrin [161], a critical protein for maintaining podocyte integrity. In addition to those findings, a role of CD80 (or B7-1), a molecule that is expressed by antigen-presenting cells upon activation and by podocytes [162], in MCD has been demonstrated. Thus, higher levels of urinary CD80 were observed in a subset of patients with MCD in relapse [163], and the presence of CD80 was noted in MCD podocytes and endothelial cells lining the capillary lumen [164]. The most recent study demonstrated CD80 as a novel downstream target of β-catenin and as a key mediator of podocyte injury and glomerulosclerosis [165]. Interestingly, circulating antinephrin autoantibodies were found to be common in adult patients with MCD, which may be considered as a biomarker of the diseases and a significant contributor to the podocyte injury [166].

Therefore, the complexity of MCD pathogenesis involves an interaction of the immune system, glomerular cells and genetics, which induces a change in activity and the expression of many factors leading to abnormalities in many mechanisms. Other innate immune pathways in MCD should also be explored for the future development of new therapeutic strategies.

### 3.5. Fabry Disease

Fabry disease (FD) is a rare, inherited genetic disorder caused by the deficient activity of alfa galactosidase A (GLA), which results in the excessive deposition of lipids in the tissue and is characterized by chronic low-grade systemic inflammation. The renal manifestation of FD occurs early in life [167,168] and results in mild proteinuria and the presence of globotriasylceramide (Gb3) in the urine between 4 and 16 years of age [169]. The deposition of glycosphingolipids occurs primarily in the cells of renal blood vessel walls and, to a lesser extent, in glomerular and tubular epithelial cells. Podocytes and distal tubular epithelial cells are shown to have the highest Gb3 levels [170], where experimental Gb3 overload results in increased integrins αν and β3 mRNA expression in cultured human podocytes at early time points and in the upregulation of CD80, TGFβ1 and CD74 inflammatory regulators at later time points [171]. Therefore, Gb3 deposition in FD seems to be one of the main (or even the main) triggers leading to pathogenic cascades associated with chronic inflammation in the kidney. However, a recent study demonstrated that the clearance of Gb3 in FD using enzyme replacement therapy failed to restore autophagy and Notch1 signaling [172], leaving Gb3 deposition as the main contributor to the FD pathogenesis in question.

Interestingly, in glomeruli with advanced lesions, immunoglobulin M (IgM) [173] and complement components (C3a, C5a and C1q) [174,175] may be presented in mesangial regions with a segmental distribution and glomerular pattern. One case report showed that FD may coexist with IgA nephropathy and dilated cardiomyopathy in the *TTN* and *BAG3* genes [175]. Interestingly, a mouse model of FD, generated by the deletion of the GLA gene and the overexpression of the Gb3 synthase, showed polyuria and renal dysfunction without remarkable glomerular damage [176], suggesting that tubular cells were affected rather than podocytes or endothelial glomerular cells (reviewed in [177]). Unexpectedly, a new study using a mouse model of FD demonstrated that diabetic FD mice have significantly higher levels of F4/80 macrophage infiltration and a lower expression of PGC1a and TFEB genes expression in the kidney compared to wildtype diabetic mice [178]. Urinary proteomics also revealed the presence of high levels of many inflammatory markers, including uromodulin, prostaglandins, podocalyxin or FGF23 [179,180,181]. Importantly, in GLA-deficient podocytes in an in vitro model of FD, the accumulation of α-synuclein (SNCA) was found to be the key mechanism of podocyte injury, where enzyme replacement therapy reduced Gb3 accumulation without the resolution of lysosomal dysfunction [182].

Therefore, further studies are required to elucidate the mechanisms of FD manifestation and progression with a focus on innate immunity, which will eventually result in the discovery of new therapeutic approaches to diagnosing and treating FD.

### 3.6. Focal Segmental Glomerulosclerosis (FSGS)

FSGS is another glomerular disease where the partial sclerosis of some glomeruli is observed, while each affected glomerulus can show only segmental involvement. FSGS was considered to have a low incidence, but in recent decades. it showed a gradual increase from 0.2 to 1.8 cases per 100,000 people [183]. Clinically, FSGS is characterized by the presence of proteinuria and nephrotic syndrome, with or without kidney function impairment (reviewed in [184]). Previously, forms of FSGS have been classified into primary (idiopathic), genetic or secondary categories (reviewed in [185]). Recently, a fourth category of “FSGS of undetermined cause” has been proposed [186]. While the mechanisms underlying FSGS pathogenesis are not completely understood, immune system dysfunction has been linked to FSGS development.

Traditionally, immune system involvement in FSGS is associated with the adaptive immune system and T-cells (reviewed elsewhere and in [187]). Several recent studies suggest the existence of circulating permeability factors that alter glomerular permeability and may be responsible for the primary FSGS form [188,189]. Thus, cardiotrophin-like cytokine factor-1 (CLCF-1), a B-cell stimulating cytokine from the IL-6 family, is present at a high concentration in the plasma of FSGS patients, with a correlation between CLCF-1 and proteinuria and lipid levels [190], and has been shown to increase glomerular permeability to albumin in vitro [191]. However, some studies do not support the idea that CLCF-1 can be the cause of primary FSGS [192]. Further, soluble urokinase plasminogen activator receptor (suPAR) is accepted as a biomarker of FSGS and is increased in most patients with FSGS [193]. These circulating markers can damage podocytes and result in the release of PRRs that are further recognized by TLRs as DAMPs. In a rat model of FSGS using 5/6 nephrectomy, the inhibition of TLR2 and TLR 4 with the small-molecule lecinoxoids VB-201 and VB-703, respectively, is sufficient to improve FSGS outcomes and reduces the number of glomerular and interstitial monocytes [194]. Furthermore, a case report from a patient with two APOL1 risk alleles demonstrated increased STING expression and the presence of collapsing glomerulopathy [195]. Interestingly, G2 APOL1 risk variant mice (1) with the genetic deletion of NLRP3, gasdermin D and STING or (2) in response to treatment with appropriate inhibitors (MCC950, di sulfiram and C176, respectively) showed lower albuminuria and improved renal function [196]. Increased NLRP3 mRNA expression in the glomeruli of FSGS patients was also demonstrated [197]. Similarly, another study using kidney tissues from FSGS patients and Adriamycin-induced mice demonstrated increased podocyte NLRP3 expression in association with reduced levels of TMEM30 [198].

This evidence suggests that innate immunity and adaptive immunity play a significant role in FSGS pathogenesis. However, there is an urgent need to study the mechanisms in more detail to foster the development of new biomarkers and treatment strategies.

## 4. Conclusions

The evidence of inflammatory pathway involvement in the pathogenesis of glomerular diseases clearly presents a need to further explore these mechanisms. Data from experimental and clinical studies demonstrate a link between various pro-inflammatory pathways, including TLR, RIG-I, CLRs, STING signaling and inflammasomes, and the pathogenesis of glomerular diseases of metabolic, non-metabolic or genetic origins (Figure 1). While the growing understanding of the innate immunity cascade and its role in glomerular damage is presented, many questions remain to be addressed. The complex crosstalk between innate and adaptive immunity remains a challenge. Why is the pathogenesis of glomerular diseases associated with the activation of chronic inflammation, while other chronic diseases do not act in this manner? How can the devastating renal inflammatory process be controlled without interfering with the normal function of the innate and adaptive immune system? Given the involvement of the inflammatory molecular pathways, how can we effectively manage inflammation in patients with different glomerular diseases? Therefore, we believe that this research topic is not yet fully explored and will yield new findings, guiding nephrologists toward new therapeutic avenues.

## Figures and Tables

**Figure 1 cells-13-01157-f001:**
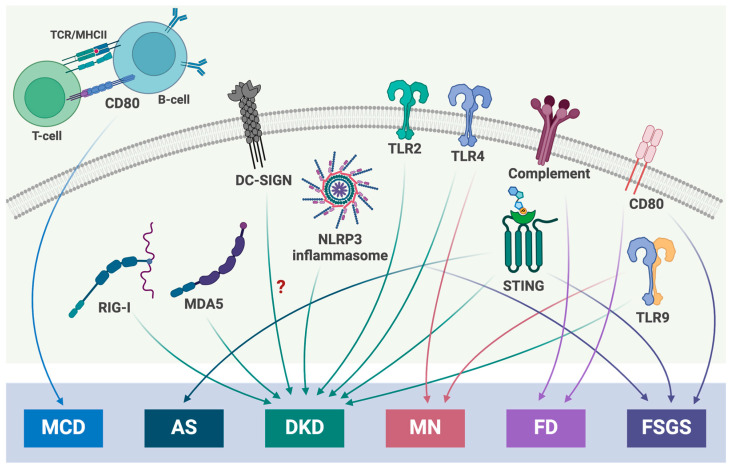
Contribution of the innate immune system to glomerular disease pathogenesis. Current knowledge suggests that many molecules of innate immunity are involved in the pathogenesis of glomerular diseases. AS—Alport Syndrome; CD80—cluster of differentiation 80; DC-SIGN—dendric-cell-specific intracellular adhesion molecule-3-grabbing non-integrin; DKD—diabetic kidney disease; FD—Fabry disease; FSGS—focal segmental glomerulosclerosis; MCD—minimal change disease; MDA5—melanoma differentiation-associated gene 5; MN—membranous nephropathy; NLRP3—NLR family pyrin domain-containing 3; RIG-I—retinoic acid-inducible gene I; STING—stimulator of interferon genes; TLR—toll-like receptor.

**Table 1 cells-13-01157-t001:** Podocyte PRRs and associated PAMPs/DAMPs activators in podocytes and associated glomerular disease.

PRRs	PAMPs/DAMPs	Downstream Effector	Glomerular Disease
DC-SIGN	Mannose	TLRs; NF-κB; INF1β	DKD?
MDA5	mtRNA	INF1β	DKD
NLRP3	mtDNA; oxmtDNA; ROS; high glucose	Caspase1; IL1β; IL-18; P2XR	DKD; FSGS
RIG-I	mtRNA	INF1β	DKD
STING	dsDNA; mtDNA	IL6; INF1β; TNFα; NF-κB	AS; DKD
TLR2	Porins; HA protein; tGPI mucin; PGN	MyD88; TIRAP	DKD; FSGS
TLR4	dsDNA; LPS	MyD88; TIRAP; TRIF; TRAM (TIR)	DKD; MN; FSGS
TLR9	dsDNA; HMGB1	MyD88; NF-κB	AS; DKD; MN

Abbreviations: dsDNA—double-strained DNA; IL—interleukin; INF—interferon; LPS—lipopolysaccharide; MyD88—myeloid differentiation primary response 88; mtDNA—mitochondrial DNA; NF-κB—nuclear factor kappa B; oxmtDNA—oxidized mtDNA; PGN—peptidoglycan; ROS—reactive oxygen species; tGPI-mucin—glycosylphosphatidylinositol-anchored mucin-like glycoproteins; TIR—Toll/interleukin-1 receptor; TIRAP—TIR domain-containing adaptor protein; TNF—tumor necrosis factor; TRAM—TRIF-related adaptor molecule; TRIF—TIR domain-containing adapter-inducing interferon-β.

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
