# Peer review of "Role of the Innate Immune Response in Glomerular Disease Pathogenesis: Focus on Podocytes"

_cells, 2024, doi:10.3390/cells13131157_

Round 1

Reviewer 1 Report

Comments and Suggestions for Authors

Dear Author,

I read your work with interest and find it generally well-written and engaging. However, I believe the focus of this work is not the role of podocytes in glomerulopathy, but rather the role of innate immunity in glomerulopathy (as also demonstrated in the figure). In fact, podocytes are mentioned only marginally in the various paragraphs. Therefore, in my opinion, either the focus of the work should be explicitly changed—along with the title, abstract, and text—or if you want to insist on the role of podocytes, the work should be completely revised to highlight and explain in more detail the specific data on podocytes. Additionally, in this case, it would be useful to add a general paragraph on podocytes and a paragraph or section on acquired immunity (B and T cells, whose alteration can have a direct impact on the podocytes themselves).

Author Response

We thank the Reviewer for these suggestions.

Unfortunately, there is very limited information on the exact role of innate immunity in podocytes since this area of research has just begun. In this review, we have attempted to compile all current knowledge related to this topic. Additionally, we have reviewed our current understanding of how PRRs in glomerular diseases, which were not initially associated with immune responses, actually have immune abnormalities contributing to their pathogenesis. Therefore, we suggest a new title for the manuscript: “Role of the Innate Immune Response in Glomerular Disease Pathogenesis: Focus on Podocytes,” which more accurately reflects the content.

While we agree with the Reviewer that B and T cells have an important impact on podocytes in the context of glomerular disease pathogenesis, our goal here is to focus solely on innate immunity. We may consider reviewing this important topic in another manuscript.

Reviewer 2 Report

Comments and Suggestions for Authors

The manuscript is very interesting and is well systematized. I have only minor suggestions:

1. Podocytes could be involved in the pathogenesis of other glomerular diseases as well (like IgA nephropathy, lupus nephritis, Fabry disease, for example). The authors might consider to discuss these as well.

2. A summarizing table would be helpful for the readers.

Author Response

The manuscript is very interesting and is well systematized. I have only minor suggestions:

  1. Podocytes could be involved in the pathogenesis of other glomerular diseases as well (like IgA nephropathy, lupus nephritis, Fabry disease, for example). The authors might consider to discuss these as well.

Reply.

In this review, our goal was to highlight the current knowledge on the role of innate immunity in pathogenesis of other glomerular diseases, which were not initially associated with immune system abnormalities. The role of the immune system in lupus nephritis and IgA nephropathy is very well characterized and a lot of elegant and detailed reviews have been published. We have added a clear statement related to this concern together with references (lines 44-45). We have added information on the Fabry disease, Section 3.5, lines 409-445. Figure 1 was revised accordingly.

  1. A summarizing table would be helpful for the readers.

Reply.

A summarizing table was added to the revised manuscript.

Reviewer 3 Report

Comments and Suggestions for Authors

The review about the role of podocyte in the immune response of glomerular disease is good and well-written. It is also an important review in this field of research. Only some comments are listed as below: 

1. In the Introduction, line 26: The glomerulus is a specialized unit with the kidney which functions as avital filtration system..... It should be a vital filtration system.

2. In the introduction, line 33: who ultimately develop end-stage kidney disease, requiring kidney transplant [1]. .... Should be requiring kidney transplant and dialysis ? 

3. In section 3.4 Minimal Change Disease (MCD), recent a new study about anti-nephrin autoantibody was published in NEJM: DOI: 10.1056/NEJMoa2314471. I suggest to add this important reference in this review. 

Minor revision is suggested to this manuscript.

Comments on the Quality of English Language

I think the quality of English language is okay. 

Author Response

The review about the role of podocyte in the immune response of glomerular disease is good and well-written. It is also an important review in this field of research. Only some comments are listed as below: 

  1. In the Introduction, line 26: The glomerulus is a specialized unit with the kidney which functions as avital filtration system..... It should be a vital filtration system.

Reply.

Corrected. Thank you.

  1. In the introduction, line 33: who ultimately develop end-stage kidney disease, requiring kidney transplant [1]. .... Should be requiring kidney transplant and dialysis ? 

Reply.

Corrected (line 35). Thank you.

  1. In section 3.4 Minimal Change Disease (MCD), recent a new study about anti-nephrin autoantibody was published in NEJM: DOI: 10.1056/NEJMoa2314471. I suggest to add this important reference in this review. 

Reply.

We thank the Reviewer for mentioning this important work. This reference was added to the revised version of the manuscript. Lines 401-404.

Round 2

Reviewer 1 Report

Comments and Suggestions for Authors

Dear Authors,

I understand your point of view and agree that it is important to emphasize how this work primarily deals with innate immunity. However, I believe the paper needs to be revised to reduce the similarity with previous works (currently 42%).

Author Response

Dear Reviewer,

Thank you for this comment.

After reviewing the iThenticate report, a new version of the manuscript has been submitted. We would like to point out that many of the highlighted items in the iThenticate report are related to molecule names, signaling pathway names, disease names, enumeration of molecular sub-families with related members, standard expressions such as '...in tubular epithelial cells in a mouse model of...' or to simple words like "is", "been", "of". Those items do not have synonyms and cannot be replaced.